# A Novel and Practicable Approach for Determining the Beam Parameters of Soft Pneumatic Multi-Chamber Bending Actuators

Frederik Lamping and Kristin M. de Payrebrune *

Institute for Computational Physics in Engineering, RPTU Kaiserslautern-Landau,
67663 Kaiserslautern, Germany
* Correspondence: kristin.payrebrune@rptu.de

**Abstract:** The design of many pneumatic soft actuators is based on multiple chambers in parallel alignment. The Cosserat beam theory is an established technique for modeling this kind of actuator, where existing approaches mainly differ in the parameters being required for simulation. The modeling approach presented in this study particularly aims at finding the beam parameters necessary for a simulation even with limited experimental methods. Importantly, it provides a straightforward relationship between the bending stiffness, the extensional stiffness and the axial stretch of the actuator. If the actuator to be modeled has an elementary design, axial measurements are sufficient to identify the parameters to perform three-dimensional simulations, which is of interest to adopters with limited testing equipment. The experimentally parameterized model of such an actuator of elementary design shows high accuracy. Both without load and with a weight of 1 N applied to the tip, the mean error of the tip position in vertical orientation is less than 3.4% for a constant extensional stiffness and less than 2.7% for a pressure-dependent extensional stiffness. Further reduction of the error could be achieved by more refined identification techniques that decompose the complex interrelationship of pressurization, forces and material stiffness.

**Keywords:** soft robotics; bending actuator; Cosserat; beam model; parameter identification

## 1. Introduction

The inherent flexibility of soft robots is advantageous in many scenarios but also a challenge for modeling. From the huge variety of soft robotic designs [1], this study focuses on the popular group of pneumatic multi-chamber bending actuators [2–6]. Sadati et al. [7] compare different modeling approaches for this kind of actuator, from which the Cosserat beam theory is identified as the most accurate "classical" one. Likewise, the Cosserat beam theory is a common approach in the literature [3,8–12]; while the mechanics of the approaches are basically similar, they differ in the parameters being required in the model, and thus the procedure to identify the beam parameters, ranging from a combination of mechanics and experiments [3] to purely data driven [12].

In this study, we present a practicable approach to determine the beam parameters of a multi-chamber actuator. This approach has two advantages: (i) through carefully chosen assumptions, it introduces a straightforward relationship between extensional stiffness and bending stiffness of multi-chamber bending actuators, including a dependency on the axial stretch, and (ii) thereby allows for accurate simulation with only limited experimental methods for parameter identification, with axial measurements being sufficient to perform three-dimensional simulations in many cases. As a side effect, designing benefits as well, since with straightforward relationships in the model estimating the influence of design changes on an actuator's characteristics becomes possible.

The model approach to determine the beam parameters is introduced in Section 2, followed by the methods of the validation process in Section 3. The results are presented in Section 4 and are discussed in Section 5.

## 2. Model

In soft robotics, beam models are often used to reduce computational costs. A beam model transforms a slender three-dimensional object, the actuator in our case, into a one-dimensional beam. The geometry and the material of the actuator are considered by beam parameters. In order to be able to simulate the deformation of the beam, these parameters need to be identified.

### 2.1. Cosserat Beam

As implied in the introduction, the Cosserat beam is an established modeling approach in soft robotics. It extends the Euler–Bernoulli beam to include large deformations, axial stretch, twist and shear, which is necessary in order to cover the full range of deformations of pneumatic soft robots. This brief introduction to the Cosserat beam theory is based on Antman [13] and Cao et al. [14].

Referring to Figure 1, the beam is represented by its centerline

$$\mathbf{q}(s) = x(s)\mathbf{e}_1 + y(s)\mathbf{e}_2 + (s + z(s))\mathbf{e}_3, \tag{1}$$

where $s$ is the arc-length parameter and $\{\mathbf{e}_1, \mathbf{e}_2, \mathbf{e}_3\}$ is a global coordinate system. The global coordinate system is supplemented by a local coordinate system $\{\mathbf{d}_1, \mathbf{d}_2, \mathbf{d}_3\}$, so-called directors, where $\mathbf{d}_1$ and $\mathbf{d}_2$ are the principal axes of inertia of the beam and

$$\mathbf{d}_3(s) = \mathbf{d}_1(s) \times \mathbf{d}_2(s). \tag{2}$$

An arbitrary vector can be transformed between the two coordinate systems by three successive rotations.

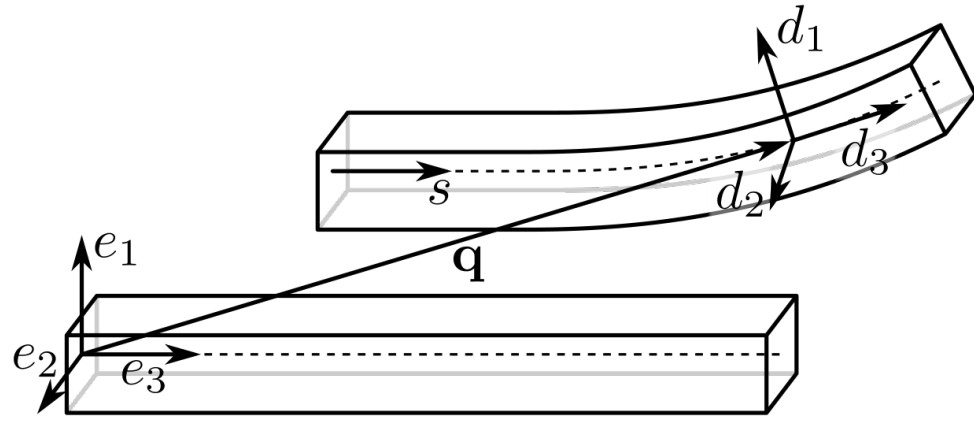

**Figure 1.** Coordinate systems of the Cosserat beam (based on Cao et al. [14]).

The static deformation of the beam results from the equilibrium of forces and moments

$$\mathbf{n}'(s) + \mathbf{f}(s) = \mathbf{0}, \tag{3}$$

$$\mathbf{m}'(s) + \mathbf{q}'(s) \times \mathbf{n}(s) + \mathbf{l}(s) = \mathbf{0}, \tag{4}$$

where $\mathbf{f}$ (force) and $\mathbf{l}$ (moment) are external line loads, $\mathbf{n}$ is the contact force (shear $n_{1/2}$, stretch $n_3$), and $\mathbf{m}$ is the contact moment (bending $m_{1/2}$, torsion $m_3$). The superscript $(.)'$ denotes the derivation with respect to the arc-length parameter $s$.

The magnitude of deformation is measured by $\mathbf{v}$ (shear $v_{1/2}$, axial stretch $v_3$) and $\mathbf{u}$ (curvature $u_{1/2}$, torsion $u_3$), where

$$\mathbf{v}(s) = \mathbf{q}(s)', \tag{5}$$

$$\mathbf{d}(s)' = \mathbf{u}(s) \times \mathbf{d}(s). \tag{6}$$

The contact forces and moments result from

$$\mathbf{n} = \mathbf{K}(\mathbf{v} - \mathbf{v}_0), \tag{7}$$

$$\mathbf{m} = \mathbf{J}(\mathbf{u} - \mathbf{u}_0). \tag{8}$$

The subscript $(.)_0$ denotes the initial position of the beam, with $\mathbf{v}_0 = (0,0,1)$ and $\mathbf{u}_0 = \mathbf{0}$ representing a straight beam in the $\mathbf{e}_3$-direction. Assuming linear-elastic material behavior,

$$\mathbf{K} = K_{ij}(\mathbf{d}_i \otimes \mathbf{d}_j) \tag{9}$$

$$K_{ij} = \begin{bmatrix} S & 0 & 0 \\ 0 & S & 0 \\ 0 & 0 & E \end{bmatrix},$$

where $S$ is the shear stiffness and $E$ is the extensional stiffness. Analogously,

$$\mathbf{J} = J_{ij}(\mathbf{d}_i \otimes \mathbf{d}_j) \tag{10}$$

$$J_{ij} = \begin{bmatrix} B & 0 & 0 \\ 0 & B & 0 \\ 0 & 0 & T \end{bmatrix},$$

where $B$ is the bending stiffness and $T$ is the torsional stiffness. For ease of reading, the stiffnesses $S$, $E$, $B$ and $T$ are combined quantities which comprise material stiffness as well as geometrical aspects, e.g., the unit of the extensional stiffness $E$ is N and the unit of the bending stiffness $B$ is Nm$^2$. Similar to others [3,8–11], in this study we restrict ourselves to extensional stiffness $E$ and bending stiffness $B$ when determining beam parameters and performing simulations. The shear stiffness $S$ and the torsional stiffness $T$ are considered infinite.

An important aspect being related to pneumatic soft robots is the consideration of pressurization in the beam model. One opportunity is to assume an initial deformation induced by pressure [8,10]. Another opportunity used by many [3,9,12,15,16], and also chosen in this study is to modify the equilibrium of forces and moments

$$(\mathbf{n} - \mathbf{F})'(s) + \mathbf{f}(s) = \mathbf{0} \tag{11}$$

$$\mathbf{m}'(s) + \mathbf{q}'(s) \times \mathbf{n}(s) + \mathbf{M}'(s) + \mathbf{q}'(s) \times \mathbf{F}(s) + \mathbf{l}(s) = \mathbf{0}, \tag{12}$$

and to apply counteracting jumping conditions for $\mathbf{F}$ and $\mathbf{M}$ at the ends of a pressurized section. Here, $\mathbf{F} = F\mathbf{d}_3$ is the axial force induced by pressure and $\mathbf{M} = M_1\mathbf{d}_1 + M_2\mathbf{d}_2$ are the corresponding moments.

### 2.2. Basic Assumptions

The first basic assumption of our novel approach is to partition the cross-section of a multi-chamber actuator with the initial length $L_0$ into areas whose extensional stiffness and dimensions are known or can be determined experimentally. Figure 2 illustrates the partitioning, where the cross-section shown has three chambers and is oriented on typical multi-chamber actuators such as the *STIFF-FLOP* [4], and the designs used by Bartholdt et al. [12] and Till and Rucker [9]. In the case of three chambers, the cross-section is partitioned in seven sections: one central, one for each chamber and one between each two chambers. More generally, with this scheme, the number of sections is $2j + 1$ for an actuator with $j$ chambers.

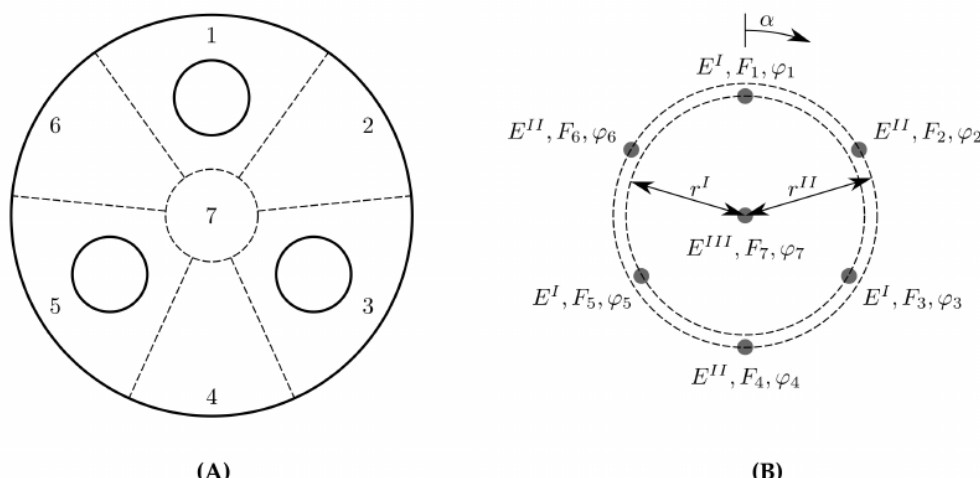

**Figure 2.** Multi-chamber actuator with three chambers embedded (**A**) and partitioning for the parameter derivation (**B**).

Each section $k$ has an extensional stiffness

$$E_k = \left\{ E^I, E^{II}, E^I, \ldots, E^I, E^{II}, E^{III} \right\}, \tag{13}$$

which is similar for sections with similar cross-sections. Internal forces $F_k$ can act on the actuator in the axial direction, being introduced to represent pressurization of the chambers, but could be used otherwise as well. In case they represent pressurization, the internal forces of the sections without a chamber are zero. Lastly, we assign a "phase angle" $\varphi_k$ to each of the sections and a circle around the centerline of radius $r_k$ on which the center points of the sections are arranged

$$\varphi_k = \left\{ 0, \frac{\pi}{n}, \frac{2\pi}{n}, \frac{3\pi}{n}, \ldots, \left( \pi - \frac{\pi}{n} \right), 0 \right\}, \tag{14}$$

$$r_k = \left\{ r^I, r^{II}, r^I, \ldots, r^I, r^{II}, 0 \right\}. \tag{15}$$

In all vectors, the last entry describes the central section.

In the following, we examine an actuator with no external loads applied that deforms with a constant curvature $\kappa$. Constant curvature is valid in this case, since it is well known that the accuracy of constant curvature models [17] only suffers from external loads. Importantly, and in opposition to these models, assuming constant curvature is only necessary in order to derive the parameters of the beam but not when they are applied in the beam model.

The second basic assumption is that the bending stiffness of the individual sections of a multi-chamber actuator is negligible compared to their extensional stiffness. This can be shown by a minor extension of the single-chamber model from a preliminary work [18] to include not only axial stretch but also bending.

### 2.3. Extensional Stiffness

Due to the constant curvature $\kappa$, the length of the individual sections $L_k$ can be determined as a function of the length of the centerline of the actuator $L$ and the direction of bending $\alpha$

$$L_k = (1 + r_k \cos(\alpha - \varphi_k)\kappa) \cdot L. \tag{16}$$

The extensional stiffness $E_k$ and the stretch ratio $\frac{L_k}{L_0}$ induce an axial force $F_{m,k}$ of the material of each individual section

$$F_{m,k} = \left( \frac{L_k}{L_0} - 1 \right) \cdot E_k. \tag{17}$$

Without external forces applied to the actuator except $F_k$, the equilibrium of axial forces at every position along the centerline of the actuator is

$$\sum_{k=1}^{2j+1} F_k - F_{m,k} = 0. \tag{18}$$

Substituting Equations (16)–(18) yields

$$\sum_{k=1}^{2j+1} F_k - \left( \frac{(1 + r_k \cos(\alpha - \varphi_k)\kappa) \cdot L}{L_0} - 1 \right) \cdot E_k = 0. \tag{19}$$

The regular arrangement of sections in a circle allows the simplification

$$\sum_{k=1}^{2j} E_k \cos(\alpha - \varphi_k) = 0, \text{if } j \geq 3, \tag{20}$$

which, in combination with $r_{2j+1} = 0$ (Equation (15)), transforms Equation (19) to

$$L = \left( \frac{\sum_{k=1}^{2j+1} F_k}{\sum_{k=1}^{2j+1} E_k} + 1 \right) \cdot L_0. \tag{21}$$

In the beam model, the extensional stiffness of the actuator $E$ is required. This parameter relates the axial force $F$ acting on the actuator to its stretch $\left( \frac{L_k}{L_0} - 1 \right)$. Equation (21) implies that

$$F = \sum_{k=1}^{2j+1} F_k, \tag{22}$$

$$E = \sum_{k=1}^{2j+1} E_k \overset{(13)}{=} j \cdot \left( E^I + E^{II} \right) + E^{III}. \tag{23}$$

As expected, the extensional stiffness of the actuator $E$ is equal to the sum of extensional stiffness of its individual sections. In particular, it is independent from the direction of bending $\alpha$ due to Equation (20).

*2.4. Bending Stiffness*

Analogous to the equilibrium of forces in Equation (18), the equilibrium of moments is

$$\sum_{k=1}^{2j+1} (F_k - F_{m,k}) \cdot \cos(\alpha - \varphi_k) \cdot r_k = 0. \tag{24}$$

Inserting Equations (16), (17) and (20), the curvature of the actuator is

$$\kappa = \frac{\sum_{k=1}^{2j+1} F_k \cdot \cos(\alpha - \varphi_k) r_k}{\sum_{k=1}^{2j+1} E_k \cdot \cos^2(\alpha - \varphi_k) r_k^2 \frac{L}{L_0}}. \tag{25}$$

The bending stiffness $B$ relates the moment $M$ acting on the actuator to its curvature $\kappa$. The moment in Equation (25) is

$$M = \sum_{k=1}^{2j+1} F_k \cdot \cos(\alpha - \varphi_k) r_k, \tag{26}$$

and thus the bending stiffness is

$$B = \frac{M}{\kappa} = \sum_{k=1}^{2j+1} E_k \cdot \cos^2(\alpha - \varphi_k) \cdot r_k^2 \cdot \frac{L}{L_0}. \tag{27}$$

Simplification is again possible because of $r_{2j+1} = 0$ (Equation (15)) and because of the regular arrangement of the sections in a circle

$$\sum_{k=1}^{2j} \cos^2(\alpha - \varphi_k) \cdot (k \mod 2) = \sum_{k=1}^{2j} \cos^2(\alpha - \varphi_k) \cdot (k+1 \mod 2) = \frac{j}{2}, \text{if } j \geq 3, \tag{28}$$

where the modulo operation distinguishes between sections with and without chamber. Hence, the bending stiffness becomes

$$B = \frac{j}{2} \cdot \left( E^I (r^I)^2 + E^{II}(r^{II})^2 \right) \cdot \frac{L}{L_0}. \tag{29}$$

Similar to the extensional stiffness $E$, the bending stiffness $B$ takes the number of sections $j$ into account, and additionally their arrangement by $r^I$ and $r^{II}$. The central section has no influence on the bending stiffness, since its length is independent from the curvature and its individual bending stiffness is neglected. In our approach, the bending stiffness results from a difference in length of the individual sections of the actuator. The more the actuator stretches due to pressurization or external loads, the larger the difference and, hence, the bending moment needs to become zero in order to remain the same curvature. Consequently, the bending stiffness increases for a higher stretch ratio $\frac{L}{L_0}$. Although the actuator is partitioned into discrete sections, the bending stiffness is independent from the direction of bending $\alpha$ because of Equation (28).

### 3. Methods

The contribution of our modeling approach is the straightforward relationship between the extensional stiffness $E$ and the bending stiffness $B$. As shown below, this is especially practicable when the robot has an elementary design: axial measurements are sufficient in order to identify all parameters necessary for three-dimensional simulations. Restriction to axial measurements is of particular interest to adopters with limited testing equipment aiming to simulate their prototypes.

The experimental validation of the model approach is conducted with an actuator, Figure 3, developed in a preliminary study [19]. Its design is modular, i.e., individual components can be adapted in order to build a custom configuration. Furthermore, the design is advantageous for modeling since the three chambers are purely cylindrical and bonded by end caps and linking elements, simplifying the kinematics. The reinforcement of the chambers consists of rings instead of a fiber, which reduces manufacturing effort but increases nonlinearity of stretching. All components except the chambers can be stiff 3D-printed or made of silicone that must be stiffer than the chambers. Figure 3 shows a configuration of the actuator with two modules stacked. Although individual pressurization of each module would be possible, throughout this study stacked chambers are supplied with the same pressure.

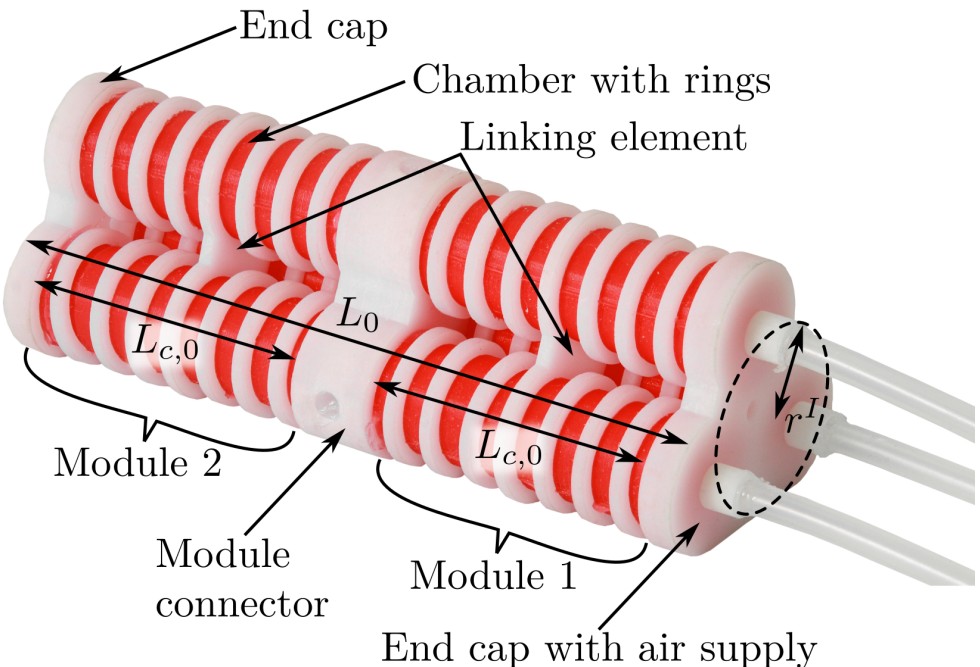

**Figure 3.** Modular three-chamber bending actuator consisting of two modules stacked with names of its components. Stiff components are white, flexible chambers are red.

Referring to Figure 4, the procedure of validating the model consists of several steps. First, the beam parameters are identified experimentally. Since two modules stacked tend to buckle when performing axial force measurements, we conducted the experimental identification in Section 3.1 with two prototypes of an individual module. For the three-dimensional measurements described in Section 3.2 and thus for validation, we used bending actuators with two modules stacked to increase the leverage of external loads. However, the axial behavior, which is investigated for the beam parameter investigation, is independent from the number of modules stacked. Conducting the beam parameter identification and the validation with different prototypes is in contrast to the usual procedure [3,8,10,12]. We chose the procedure to demonstrate that three-dimensional simulations can be performed with limited testing equipment. All actuators used in this study were poured from the same silicone, and with exactly the same time in vacuum and curing temperature.

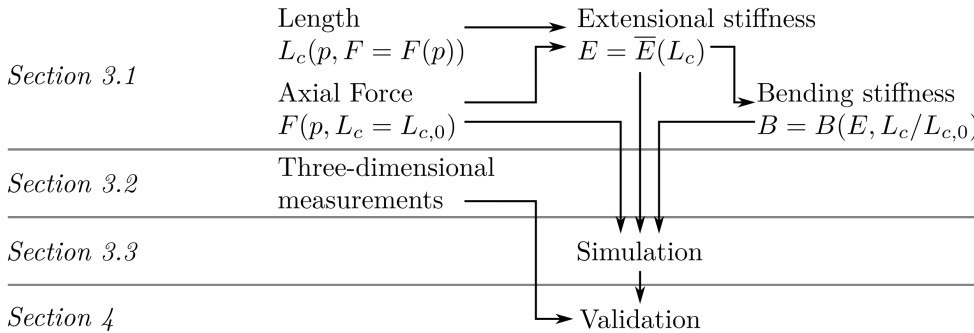

**Figure 4.** Procedure of validation with corresponding sections.

### 3.1. Experimental Parameter Identification

Comparing the actuator of investigation in Figure 3 to the partitioning scheme in Figure 2, we find that $E^{II} = E^{III} = 0$ since the actuator consists of individual chambers bonded by stiff components. In this case, Equations (23) and (29) simplify to

$$E = 3 \cdot E^I, \tag{30}$$

$$B = \frac{3}{2} \cdot E^I (r^I)^2 \cdot \frac{L_c}{L_{c,0}}. \tag{31}$$

Consequently, the bending stiffness $B$ of the actuator results from its extensional stiffness $E$ and from the radius $r^I$ of the circle on which the center points of the chambers are arranged. Independently from the actuator's design, the bending stiffness is always a function of the stretch ratio $\frac{L_c}{L_{c,0}}$. The subscript $(.)_c$ refers to the length of the chambers, which needs to be distinguished from the length of the actuator $L_0$, since the actuator also consists of stiff components which do not stretch.

We use axial measurements in order to determine the extensional stiffness $E$ of the actuator. However, the extensional stiffness of the individual chambers could also be determined by a model, as done in a preliminary study of the authors [18] and by many others [20–22]. If the actuator was more complex, as in Figure 2, the experimental parameter investigation would have to be extended. After axial measurements for the extensional stiffness $E$, one (or several) bent positions would have to be analyzed in order to determine $\left(E^I(r^I)^2 + E^{II}(r^{II})^2\right)$ from Equation (29) as a combined quantity and to determine the bending stiffness $B$.

In order to find the relationship between pressurization and axial force, we conducted axial force measurements according to Figure 5a, with the actuator fixed at its initial length ($L = L_0$ and $L_c = L_{c,0}$). We used a *VTEM* (*Festo GmbH & Co. KG, Germany*) pneumatic terminal to pressurize the chambers of the actuator (always equally) and an *S2M* (*Hottinger Brüel & Kjaer GmbH, Germany*) 50 N sensor to measure axial forces. A measurement consists of ten cycles. As shown in Figure 5b, in each cycle the pressure is increased from 0 kPa to 96 kPa and decreased back to 0 kPa in steps of 24 kPa, resting 4 s at each pressure level. In order to suppress any transient effects, we defined the mean value of the last second of each step as the force measured, and calculated the mean value of all measurements for each pressure level. For both actuators tested, pressure and force are approximately proportional to each other. The maximum force is approximately 33 N.

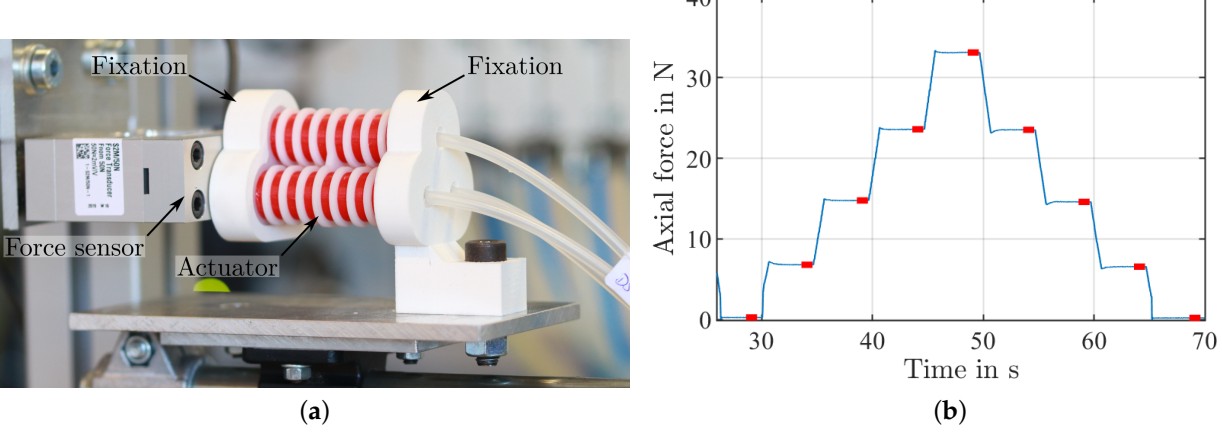

|     |     |
| --- | --- |
| (**a**) | (**b**) |

**Figure 5.** Setup of axial force measurements (**a**) and an individual cycle of a corresponding measurement, with the raw force signal in blue and the mean value for each pressure level in red (**b**).

Additionally to the axial force measurements, we measured the length of the actuators ($L$ and $L_c$) when purely pressurized (without external forces applied) with a sliding caliper

for similar steps of pressure. This method was chosen in order to consider limited testing equipment. However, we verified the results by measurements with the same setup as presented in Section 3.2. The chambers of both actuators stretch approximately 20% at maximum pressure, undergoing a progressive curve.

The extensional stiffness $E$ of the actuator is the ratio of force to stretch. Due to the progressive curve of the stretch, the extensional stiffness $E$ is not constant, where $E$ is lower for higher forces. Since the model considers a constant extensional stiffness, we used the mean value of the curve, which is approximately $E = 200\,\text{N}$ for both actuators tested.

While the identification method described is practical, it causes inaccuracies of the parameters in two ways. First, the extensional stiffness $E$ is considered constant. Second, we determine it as the relationship of length to axial force, both with respect to pressure. However, the axial force is only measured at initial length for simplification although it might differ depending on the current length of the actuator.

### 3.2. Three-Dimensional Measurements

While the aim of the parameter identification procedure was to be as simple as possible, we used a more refined setup at the match Institute, Hannover (Figure 6). Again, two actuators of the same configuration were tested. The actuators consist of two modules stacked and were clamped in vertical position, with and without a load of 1 N attached to the free end. Stacking modules increases the effect of self-weight and of the external load. The three-dimensional deflection and the orientation of the tip were captured by a *Prime 17W* (*OptiTrack*) optical motion tracking system with five markers attached to the free end of the actuators. The $z$-axis of the coordinate system was calibrated along the centerline of the actuator such that the position of the free end without load was $(0, 0, L_0)$ in the initial position.

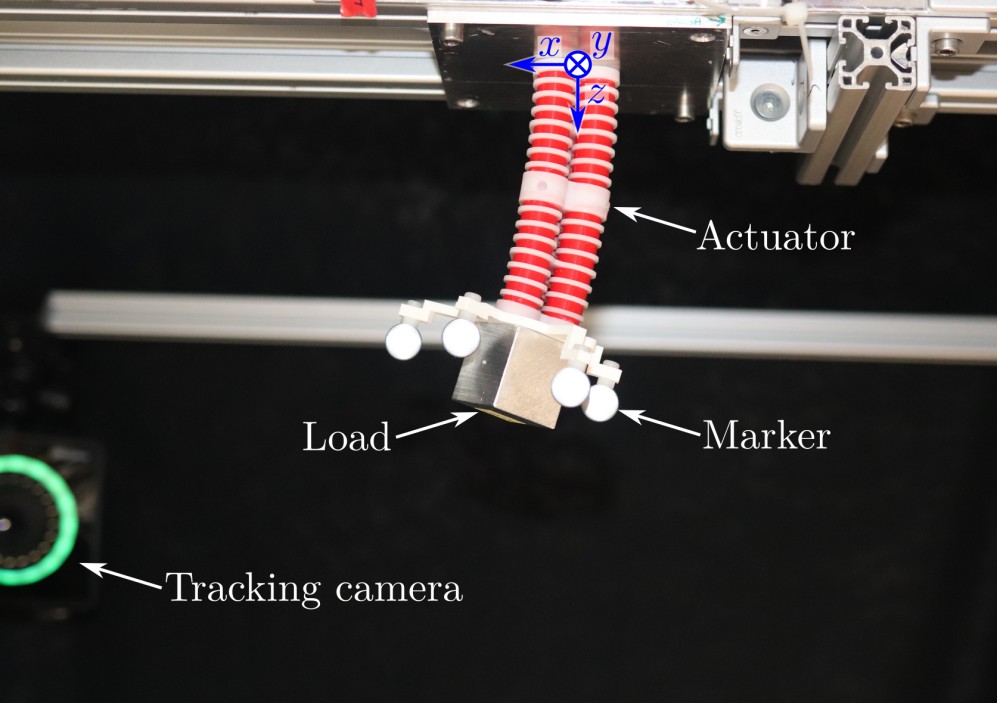

**Figure 6.** Setup of the three-dimensional measurements: actuator with markers and load attached in vertically clamped and slightly bent position.

Comparable to the identification of the extensional stiffness $E$, we varied the pressurization of the chambers from 0 kPa to 95 kPa in steps of 23.75 kPa, resting 3 s in every position and using the mean value of the last second of resting for analysis. Considering every combination of pressurization of the chambers, a measurement consisted of 125 data

points. We permuted the order of positions randomly in order to suppress any long-term changes of deflection. The data points are used for validation of the experimentally identified beam parameters, cf. Figures 7 and 8.

### 3.3. Simulation

Using the *bvp4c* solver in *Matlab R2021a*, we implemented the static case of the Cosserat beam (Equations (11) and (12)) with the parameters in Table 1. Since the actuators consist of components with different weights, the beam is divided into six regions, as shown in Table 2. Gravitational forces corresponding to the weight of individual regions are considered as a line load in the *z*-direction. Some of the regions overlap at the real actuator, e.g., the chambers are slightly inserted to the end caps. The regions are chosen such that the distribution of weight of two overlapping components combined in one region is approximately balanced. According to Equation (29), the bending stiffness depends on the axial stretch ratio $\frac{L_c}{L_{c,0}}$ of the actuator, which is represented by $v_3$ in the Cosserat beam model. We assumed infinite extensional stiffness and bending stiffness for the stiff components of the actuator, and an infinite shear stiffness and torsional stiffness for all components.

**Table 1.** Parameters of the actuators used in the simulation.

| | $L_{c,0}$ in m | $L_0$ in m | $E = 3 \cdot E^I$ in N | $E^{II}$ in N | $E^{III}$ in N | $r^I$ in m |
|---|---|---|---|---|---|---|
| Actuator 1 | $44.95 \cdot 10^{-3}$ | $105.9 \cdot 10^{-3}$ | 200 | 0 | 0 | $\frac{20}{\sqrt{3}} \cdot 10^{-3}$ |
| Actuator 2 | $44.8 \cdot 10^{-3}$ | $105.6 \cdot 10^{-3}$ | | | | |

**Table 2.** Weight and dimensions of actuators used in the simulation, from fixed end to free end (each two alternatives for regions 2, 4 and 6).

| Region | Components | Weight in kg | Length in m | Annotation |
|---|---|---|---|---|
| 1 | End cap 1 | $3.97 \cdot 10^{-3}$ | $3.5 \cdot 10^{-3}$ | |
| 2 | Chamber + rings ($L_{c,0}$) | $25 \cdot 10^{-3}$ | $44.95 \cdot 10^{-3}$ | Actuator 1 |
| | | | $44.8 \cdot 10^{-3}$ | Actuator 2 |
| 3 | Connector cap | $8.26 \cdot 10^{-3}$ | $9 \cdot 10^{-3}$ | |
| 4 | Chamber + rings ($L_{c,0}$) | $25 \cdot 10^{-3}$ | $44.95 \cdot 10^{-3}$ | Actuator 1 |
| | | | $44.8 \cdot 10^{-3}$ | Actuator 2 |
| 5 | End cap 2 + mounting plate marker | $13.97 \cdot 10^{-3}$ | $7.5 \cdot 10^{-3}$ | |
| 6 | Marker | $11.4 \cdot 10^{-3}$ | $17.2 \cdot 10^{-3}$ | No load applied |
| | Marker + load | $113.8 \cdot 10^{-3}$ | $34.9 \cdot 10^{-3}$ | Load applied |

According to Equations (11) and (12), line loads along the chambers (regions 2 and 4 of the beam, Table 2) are applied due to pressurization. Additionally, counteracting jump conditions are applied at the ends of the chambers. For every position of the three-dimensional measurements, we know the pressurization, from which we calculate the force $F_k$ induced at the individual chambers. The axial force $F$ is equal to the sum of forces $F_k$ of the individual chambers. The moments $M_1$ and $M_2$ correspond to the forces $F_k$ of the individual chambers and their distance to the centerline in the *x*-direction and *y*-direction, respectively.

The absolute error of a simulation is the Euclidean distance between the tip position in the experiment and in the simulation. The relative error is determined with two different reference lengths: the length of the actuator, as is common in the literature [3,8,10,12,23], and the deflection of the tip in the experiment [11], as a more refined reference method.

The error of the latter is higher since the deflection is typically lower than the length of the actuator.

## 4. Results

The outcome of each three-dimensional measurement is a set of 125 data points comprising the position and the orientation of the actuator tip. We tested two actuators, each with and without a weight of 1 N applied to the free end. The main aspect of the model validation is the quantitative comparison of the test data to data achieved by simulation. Additionally, we compare the shape of the actuator in reality and in simulation for two selected positions.

### 4.1. Tip Position

The results (experimental and simulated data) for both actuators tested without load are shown in Figure 7. Of all the data points (shown as unfilled circles), we consider particularly important extreme positions (shown as filled circles) at which (i) the pressure of one chamber is maximum and of the others is zero (Positions 1, 3, 5), (ii) the pressure of two chambers is maximum and of the third is zero (Positions 2, 4, 6), and (iii) the pressure of all chambers is maximum (Position 7). For these extreme positions, a comparison of the coordinates and the error are additionally shown in Figure 7.

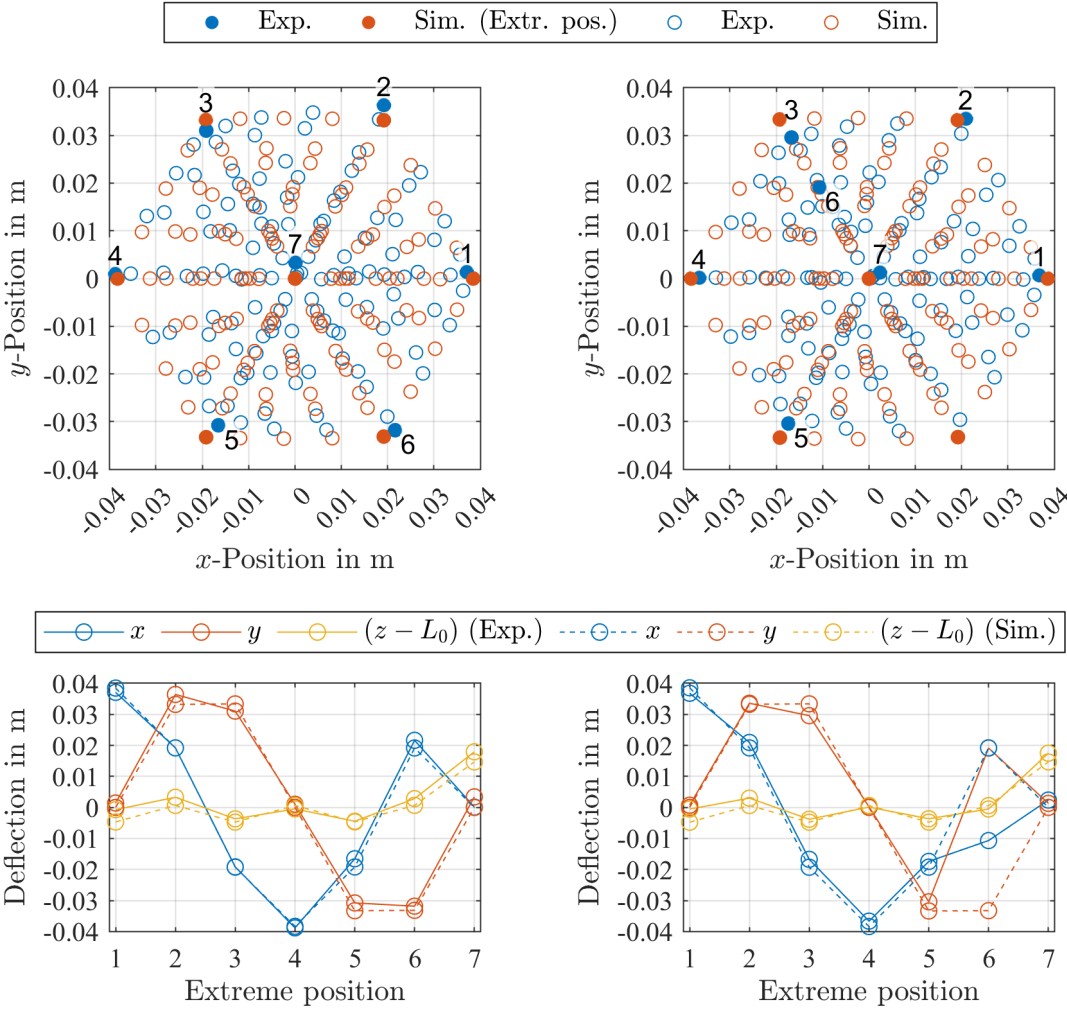

**Figure 7.** *Cont.*

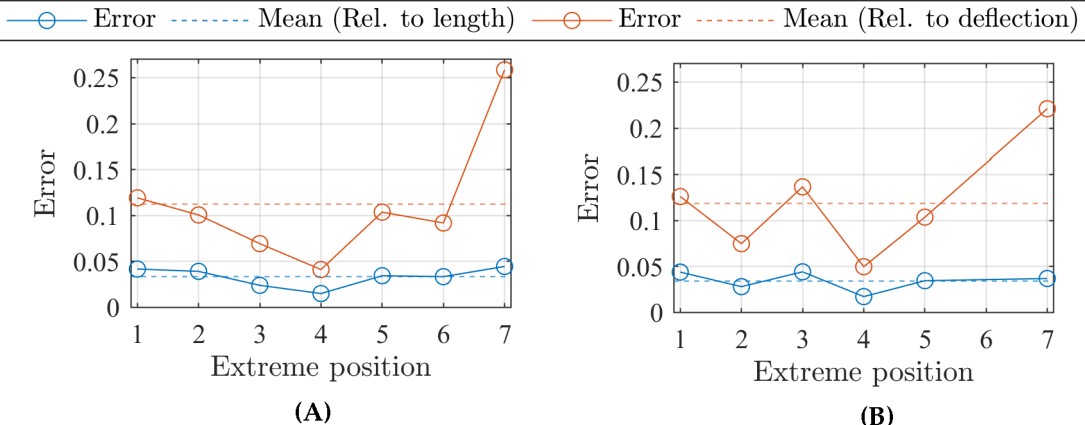

**Figure 7.** Actuator 1 (**A**) and Actuator 2 (**B**) without load: tip position in *x*–*y* plane with extreme positions highlighted (top), coordinates of seven extreme positions (middle) and error at these positions (bottom).

Throughout all measurements (also with load), the deflection of Position 2 is remarkably larger than for the other extreme positions, which also applies for the other data points in this "direction". However, a structural error in the setup could not be identified. In addition, hand manufacturing has a significant influence, as can be seen from Position 7 since the *x*-coordinates and *y*-coordinates in the experiments are not zero. Comparison of the experimental data of the two actuators results in deviations of 2.1% (mean deviation of the extreme positions relative to the actuator length).

In general, the simulation results and the experimental results without load in Figure 7 agree well. This is also depicted by the mean error of the extreme positions relative to the actuator length (3.3%/3.4%), which is in a range similar to comparable approaches [3,8,10,23,24]. Position 6 of Actuator 2 in Figure 7B is an outlier (the only one of all data points measured) and therefore not considered in the mean error. Note that restricting on the extreme positions increases rather than decreases the mean error compared to considering all data points.

The results with a load of 1 N applied to the free end of the actuator are shown Figure 8. Again, the mean error of the extreme positions (2.9 %/3.1 %) is still in a range typical for comparable approaches without load and with load [25]. However, it is significantly lower than the error of Sadati et al. [7], where also a load is applied to the tip of the actuator.

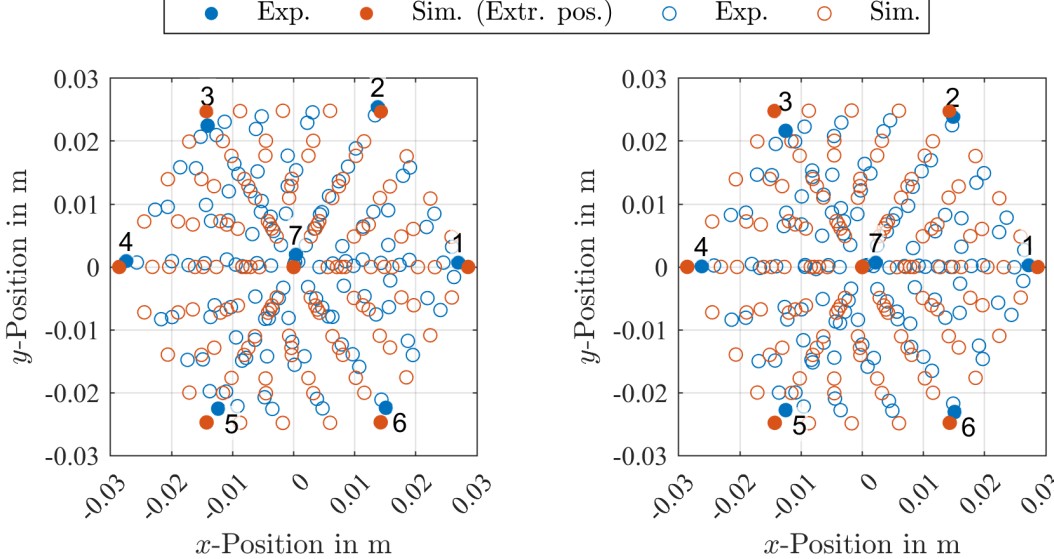

**Figure 8.** *Cont.*

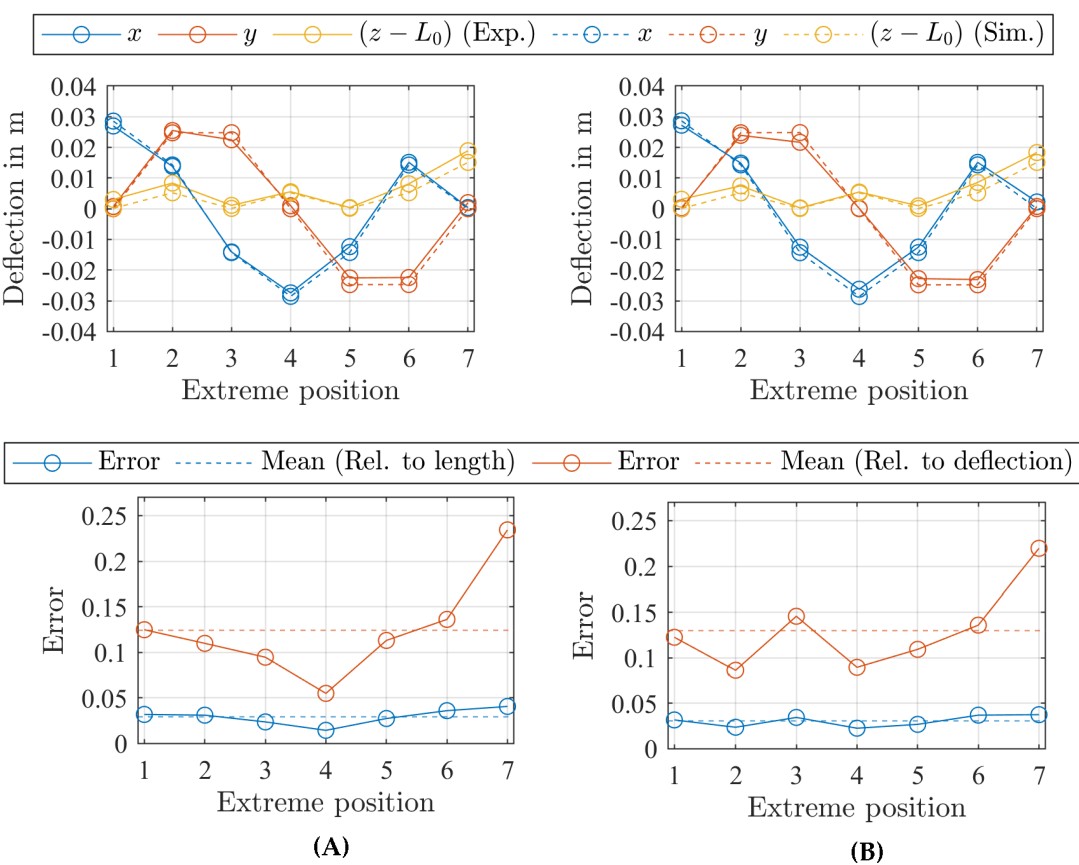

**Figure 8.** Actuator 1 (**A**) and Actuator 2 (**B**) with load: tip position in *x–y* plane with extreme positions highlighted (top), coordinates of seven extreme positions (middle) and error at these positions (bottom).

The extensional stiffness $E$ has an important impact in our approach, since it also determines the bending stiffness. For the simulations, we used $E = 200\,\mathrm{N}$ that was identified as a mean value in a simple experiment without any fitting methods. In order to verify the sensitivity of the error on $E$, we performed additional simulations with varied $E = 180\,\mathrm{N}$ and $E = 220\,\mathrm{N}$, and with a pressure-dependent extensional stiffness $E(F)$ being known from the experimental parameter identification. The results are shown in Table 3. As expected from Figures 7 and 8, where the simulation predicts too large deflections of the tip, increasing the extensional stiffness to $E = 220\,\mathrm{N}$ lowers the error compared to $E = 200\,\mathrm{N}$. Using a pressure-dependent extensional stiffness $E(F)$ lowers the error to a range close to the deviation between the prototypes due to manufacturing errors.

**Table 3.** Mean error of the extreme positions (in percent) relative to actuator length and tip deflection, respectively, for varying extensional stiffness $E$.

|  |  | $E = 180\,\mathrm{N}$ | $E = 200\,\mathrm{N}$ | $E = 220\,\mathrm{N}$ | $E = E(F)$ |
|---|---|---|---|---|---|
| Actuator 1 | 0 N | 5.0 / 15.5 | 3.3 / 11.2 | 3.6 / 12.3 | 2.7 / 8.7 |
|  | 1 N | 3.8 / 15.4 | 2.9 / 12.4 | 2.8 / 11.9 | 2.1 / 8.7 |
| Actuator 2 | 0 N | 5.6 / 18.0 | 3.4 / 11.9 | 2.7 / 10.0 | 2.0 / 7.0 |
|  | 1 N | 4.0 / 16.6 | 3.1 / 13.0 | 2.6 / 11.4 | 2.3 / 9.4 |

### 4.2. Actuator Shape

While the tip position is an important quantity to compare the accuracy of our model to others, the shape of the actuator is a qualitative measure that provides further insight

into the model. Figure 9A shows the shape of Actuator 2 in Position 1, as well as the corresponding simulation results (centerline of the actuator solid and parallel offset curve dashed) for an extensional stiffness $E = 200\,\mathrm{N}$. While the shape of the actuator without load is well predicted by the simulations, the axial stretch of the actuator is predicted too large, which is in agreement with Figure 7B. With load applied, the deflection of the tip is larger in the simulation than in the experiment, in agreement with Figure 8B. Referring to the Discussion (Section 5), we consider the complex interrelation of pressure, forces and material stiffness to be the reason.

In Figure 9B, the actuator is turned by 90 deg such that the clamping is horizontal. In opposition to the vertical Position 1, the simulation predicts a larger deflection than in the experiment. This is caused by the shear modulus being assumed to be infinite in the simulation. The rings reinforcing the chambers of the fixed half of the actuator indicate that the cross-section is not perpendicular to the centerline of the actuator, which is an indication of shear.

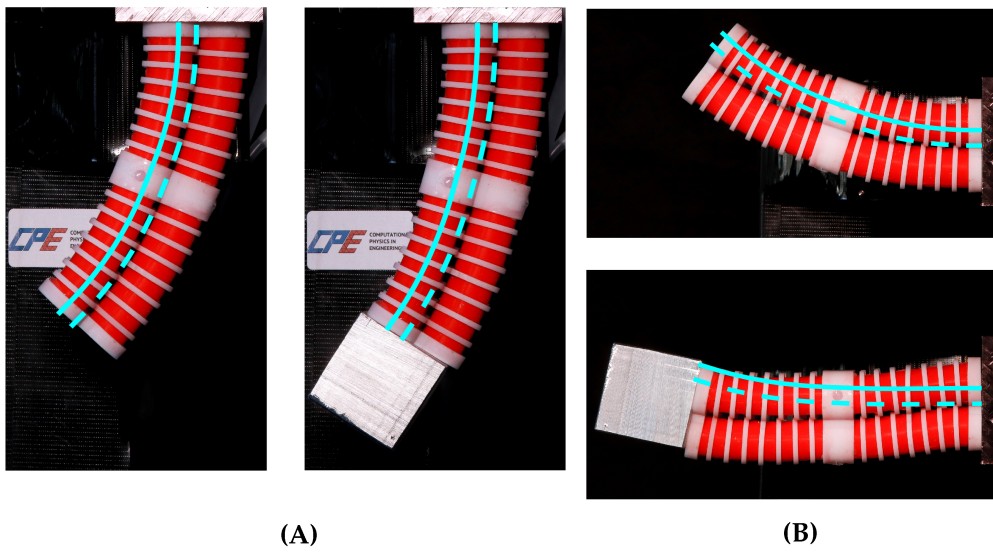

**(A)**             **(B)**

**Figure 9.** Actuator 2 (without markers) with an individual chamber at maximum pressurization in vertical position (**A**) and horizontal position (**B**), both with and without load applied. The simulated results are presented in two ways: the centerline of the actuator (solid) and, for better comparison of the results, an offset curve [26] of the centerline between the two chambers visible (dashed).

## 5. Discussion

The novelty of the modeling approach presented in this study is the practicability of finding the beam parameters of a multi-chamber bending actuatorand that the bending stiffness depends on the axial stretch of the actuator. If the actuator to be modeled has an elementary design, axial measurements are enough to obtain sufficient model parameters for accurate three-dimensional simulations, which is demonstrated by the results in Section 4. The model approach is of advantage for adopters with limited testing equipment and provides the possibility of estimating the influence of design changes on the deformation characteristics of the actuator. However, from the authors' perspective, the results imply another two important aspects.

First, experimental investigation of soft robots suffers from manufacturing errors. Hence, transferring parameters determined from one robot to another, as is done in this study, is always limited. For example, the deviation between the extreme positions of the two actuators tested is 2.1% relative to the actuator length, which means that simulations can never be reliably below this limit. This problem is well-known, and typically addressed in the literature by using the same robot for determination of parameters and validation of the model [3,8,10,12].

Second, the interrelation of pressurization, forces induced by pressure and material stiffness is complex. For example, while the simulations without external load predict the shape of the actuator well, the predicted shape deviates from the experimental results with a load applied to the tip of the actuator. Besides possible inaccuracies in the model itself, this is caused by errors in the experimentally identified relationship between pressurization and force, since the relationship is determined at the initial length but is also applied to the stretched actuator. This error also has an influence on the extensional stiffness and the bending stiffness. As long as the forces induced by pressure dominate external loads, the error is eliminated in the simulation, because the stiffness and the dominating forces both include the same error. For higher external loads, the error is not eliminated because it is included in the stiffness but only partially in the loads.

Our future work is motivated by these aspects. Bartholdt et al. [12] show that hybrid modeling (combination of classical modeling and data based approaches) is a promising approach to take the complex interrelation of pressurization, forces and material stiffness into account. Vice versa, the results of the experimental investigation in this study suggest that underlaying the presented modeling approach with hybrid modeling would reduce the experimental data required, while keeping accuracy and increasing generality, since evaluation of the parameters determined becomes possible. Likewise, additional parameters such as the shear stiffness, which is neglected in this study and which is shown to be important in some scenarios, can be included with little effort.

**Author Contributions:** Conceptualization, F.L.; methodology, F.L.; validation, F.L.; investigation, F.L.; data curation, F.L.; writing—original draft preparation, F.L.; writing—review and editing, K.M.d.P.; visualization, F.L.; supervision, K.M.d.P.; project administration, K.M.d.P.; funding acquisition, K.M.d.P. All authors have read and agreed to the published version of the manuscript.

**Funding:** Funded by the Deutsche Forschungsgemeinschaft (DFG, German Research Foundation)–404986830–SPP2100.

**Institutional Review Board Statement:** Not applicable.

**Informed Consent Statement:** Not applicable.

**Data Availability Statement:** The data presented in this study are available on request from the corresponding author.

**Acknowledgments:** The authors would like to thank Mats Wiese from the Institute of Assembly Technology (match), Leibniz University Hannover for providing the three-dimensional test stand.

**Conflicts of Interest:** The authors declare no conflict of interest.

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
