# Peer review of "A Novel and Practicable Approach for Determining the Beam Parameters of Soft Pneumatic Multi-Chamber Bending Actuators"

_applsci, doi:10.3390/app13052822_

Round 1

Reviewer 1 Report

This paper is well written  and  the structure of the document is good.I have the following comments.

 A)    In eq.  9  operation  should be defined....

    B)    In   section  Cosserat beam  ….it would be fruitful for a potential reader to present a Figure of beam with  corresponding  parameters  (1)-(6)….

Author Response

Thank you for your review. Please see PDF for PTP-reply.

Reviewer 2 Report

The manuscript looks good and I recommend to publish this article after addressing my minor comments.

  1. What is the main design for many pneumatic soft actuators?
  2. What is the Cosserat beam theory and how does it apply to modeling this type of actuator?
  3. How do existing approaches for modeling pneumatic soft actuators differ from each other?
  4. What is the aim of the modeling approach presented in the study?
  5. How does the study's approach provide a relationship between the bending stiffness, extensional stiffness, and axial stretch of the actuator?
  6. What type of measurements are needed to identify the parameters for three-dimensional simulations?
  7. Was the experimentally parametrized model of the actuator of elementary design accurate?
  8. How well did the model perform without load and with a weight applied to the tip?
  9. Can the error in the tip position be reduced further with more refined identification techniques?
  10. What is the advantage of the inherent flexibility of soft robots?
  11. What type of soft robotic designs is the study focusing on?
  12. What was the most accurate "classical" approach identified by Sadati et al. for modeling pneumatic multichamber bending actuators?
  13. What is the advantage of the approach presented in the study for determining the beam parameters of a multi-chamber actuator?
  14. How does the approach allow for accurate simulation with limited experimental methods for parameter identification?
  15. How does the approach make designing benefits possible by estimating the influence of design changes on the actuator's characteristics?

Author Response

(The authors gave the same response as above.)

Reviewer 3 Report

In the manuscript, the authors presented a model based on Cosserat beam theory to find beam parameters for commonly studied pneumatic soft actuators containing multiple parallel aligned chambers. On the basis of some assumptions and simplifications, the soft actuator was partitioned into several sections with different modeling features and the extensional stiffness and bending stiffness of the entire actuator were derived. Followingly, experiments using a fabricated actuator were conducted to track the tip position upon pressurization with different magnitudes or to different chambers. Modeling simulation results well aligned with experimental results. Overall, the manuscript is clearly written, of good quality, and highly interesting for a broad audience. I just have minor comments.

1.       Can the authors show a table of designed parameters of their soft actuators for good readability? For example, page 7, line 180, there is no idea to suddenly show that r_I = 20/sqrt(3) mm.

2.       Page 10, line 287, the outlier seems very unreasonable. Can the authors explain why it happened?

3.       What are the authors’ thoughts in terms of modeling if gravity is also involved?

Author Response

(The authors gave the same response as above.)
